# Bioactive Electrospun Fibers of Poly(ε-Caprolactone) Incorporating α-Tocopherol for Food Packaging Applications

**DOI:** 10.3390/molecules26185498

**Published:** 2021-09-10

**Authors:** Raluca P. Dumitriu, Elena Stoleru, Geoffrey R. Mitchell, Cornelia Vasile, Mihai Brebu

**Affiliations:** 1Laboratory of Physical Chemistry of Polymers, “Petru Poni” Institute of Macromolecular Chemistry, 41A Gr. Ghica Voda Alley, 700487 Iaşi, Romania; rdumi@icmpp.ro (R.P.D.); elena.paslaru@icmpp.ro (E.S.); cvasile@icmpp.ro (C.V.); 2Centre for Rapid and Sustainable Product Development, Institute Polytechnic of Leiria, Rua de Portugal, 2430-028 Marinha Grande, Portugal; geoffrey.mitchell@ipleiria.pt

**Keywords:** electrospun fibers, hydrophobic coating, rheology, food simulant migration, food spoilage

## Abstract

Antioxidant activity is an important feature for food contact materials such as packaging, aiming to preserve freshness and retard food spoilage. Common bioactive agents are highly susceptible to various forms of degradation; therefore, protection is required to maintain functionality and bioavailability. Poly(ε-caprolactone) (PCL), a biodegradable GRAS labeled polymer, was used in this study for encapsulation of α-tocopherol antioxidant, a major component of vitamin E, in the form of electrospun fibers. Rheological properties of the fiber forming solutions, which determine the electrospinning behavior, were correlated with the properties of electrospun fibers, e.g., morphology and surface properties. Interactions through hydrogen bonds were evidenced between the two components. These have strong effect on structuration of macromolecular chains, especially at low α-tocopherol amounts, decreasing viscosity and elastic modulus. Intra-molecular interactions in PCL strengthen at high α-tocopherol amounts due to decreased solvation, allowing good structural recovery after cease of mechanical stress. Morphologically homogeneous electrospun fibers were obtained, with ~6 μm average diameter. The obtained fibers were highly hydrophobic, with fast release in 95% ethanol as alternative simulant for fatty foods. This induced good in vitro antioxidant activity and significant in vivo reduction of microbial growth on cheese, as determined by respirometry. Therefore, the electrospun fibers from PCL entrapping α-tocopherol as bioactive agent showed potential use in food packaging materials.

## 1. Introduction

The development of engineered materials with desired functionality and tailored surface properties is a permanent challenge. Research emerged in last decades with the aim to improve the quality of life through the elaboration of new biodegradable and environmental-friendly advanced polymeric materials.

Nowadays there is a continuously growing interest in food packaging, due to the change in lifestyle and eating habits. Until recently the sole purpose of packaging was to maintain the quality of food while protecting it from various hazards during the transportation and storage before consumption [1,2,3]. Traditional materials were based on fossil resources and of great environmental concern being non-biodegradable and difficult to recycle due to complex composition and high contamination [4]. Recent trends in the field of food packaging are directed towards two main important aspects. One is the necessity to provide protection and preservation, and this can be achieved through functionality that can be induced by incorporation of active principles into polymeric matrices. Another aspect is the sustainability and eco-friendly features required to save fossil resources and to decrease the waste amounts by increasing the recyclability or the biodegradability [5,6].

Food quality is strongly affected by the action of the omnipresent oxygen, which induces various spoilage processes such as lipid oxidation or microorganisms growth. Oxidation leads to degradation of proteins, vitamins or essential fatty acids, with effect in loss of flavor and color changes, decreasing the food shelf life [7]. Therefore, introduction of antioxidants into food packaging materials became an attractive solution to reduce or delay such undesired processes. Among the active agents frequently used, scientific literature reports about essential oils and vegetal extracts from different edible and medicinal plants, herbs and spices such as basil, thyme, clove, cinnamon, ginger, sage, rosemary, etc. [8,9,10,11], enzymes (e.g., glucose oxidase, lysozyme) [12,13], polysaccharides (e.g., chitosan) [14], vitamins such as vitamin E (α-tocopherol) [15,16] or vitamin C (ascorbic acid) [1,2,7].

In many cases it is desirable to modify only the surface of materials in order to ensure certain functionality, but not to change the bulk properties. Therefore, an attractive option is to incorporate the active principles into coatings [17]. Electrospinning is a versatile and low cost technique which enables the production of homogenous, continuous and uniform fibers with submicron down to nanometer diameter size [18,19]. Electrospun fiber mats and coatings can be tailored with barrier and active properties advantageous for various applications in industry (multifunctional membranes, nanocomposites, filtration systems, textiles/protective clothing, coatings) or in biomedical field (3D scaffolds for tissue engineering, wound dressings, vascular grafts, drug delivery systems, biosensors, enzyme immobilization) [20,21]. Electrospinning offers the possibility to produce nanostructured fibrous materials with various active principles entrapped for active and intelligent packaging systems. The use of different electrospinning techniques (uniaxial, co-axial, emulsion etc.) opens a wide range of possibilities for producing fibers with different morphologies (e.g., porous, core-shell, hollow, aligned, multi-layered) and their associated structural and functional advantageous properties, especially the high surface-to-volume ratio and high loading capacity, which allows a subsequent sustained/controlled release of the active compounds [22]. Electrospinning was successfully tested to obtain various polymeric materials for food packaging, either by entrapping of bioactive agents or by deposition of electrospun fibers as coatings onto polymeric films to structure the surface and to confer desired functionalities [23]. In particular, the development of new formulations by nanoencapsulation of various bioactive compounds was recently exploited for delivery of nutraceuticals in foods [24]. Recent research reports on active packaging obtained by incorporation of vanillin or quercetin as natural active agents into zein fibers prepared by combining electrospinning and supercritical fluids impregnation [25] or encapsulated in cyclodextrin inclusion complex [26]. Fiber mats were produced from co-electrospun ethanolic extract of *Hibiscus rosa sinensis* containing anthocyanin with polycaprolactone, polyethylene oxide and silver nanoparticles with the aim to be used as freshness indicator due to pH sensitivity induced color change [27].

The electrospinning behavior strongly depends on the rheological properties of the polymeric solutions, which determine the ability to spread over the substrate, the thickness and the uniformity of deposition and, consequently, the application performance of the final fibers. Decreased viscosity of polymeric solutions offers a processing advantage during high shear operations, such as pumping, filling and spraying application; however too low viscosities can interrupt the continuous jet, leading to electrospray of discontinuous droplets [16,28].

The choice of components for coating materials is largely dependent on the final desired function. Here we selected poly(ε-caprolactone) (PCL) as polymer matrix and α-tocopherol as active principle. PCL is a semi-crystalline saturated aliphatic synthetic polyester with good biocompatibility and suitable mechanical properties for specific biomedical applications such as drug delivery device, suture, adhesion barrier [29,30]. Recently, PCL gained increasing interest for food packaging applications due to its hydrophobic character, relatively good barrier properties derived from its semi-crystalline nature, biodegradability and the fact that it is approved by Food and Drug Administration (FDA) [31,32,33]. α-tocopherol is a fat-soluble phenolic compound found in plant tissues, with strong anti-inflammatory effect and good protection against UV radiation. It is the main form of vitamin E, the most chemically reactive and biologically active [34], and can be used as a strong antioxidant agent, due to its reducing activity for free radicals. It is also the preferred form of vitamin E adsorbed by humans, being considered GRAS (Generally Recognized as Safe) and authorized by EU Regulation [35]. Addition of vitamin E or of its main components (e.g., α-tocopherol) to food packaging can be useful, since, besides induced antioxidant properties, it can lead to fortified food.

We previously reported on antioxidant and photoprotective films based on alginate and lignosulfonate [36], on incorporation of rosemary powdered extracts in PLA based films [37] and on bioactivation of inert polyethylene and PLA films by coating with α-tocopherol and/or chitosan formulations [15,38], all with potential application in food packaging.

In a preliminary conference proceeding paper [39] we shortly presented few aspects on obtaining electrospun PCL/α-tocopherol fibers with promising antioxidant activity observed from DPPH assay. Here we reveal detailed information from the characterization of the fiber forming solutions and of the obtained electrospun fibers. A complex rheological study on PCL/α-tocopherol solutions points out the effect of α-tocopherol concentration on the intra- and inter-molecular interactions of PCL, which are also proved by FTIR spectroscopy. Surface morphology and topography of electrospun fibers are described in detail based on SEM and AFM microscopy. Specific properties of materials, such as surface wettability, ABTS antioxidant assay, migration tests in food simulants and respirometry tests on cheese, indicating their suitable usage for food packaging applications, are presented and discussed. A novel aspect introduced in this paper is drawing correlations between the rheology of solutions and the properties of electrospun fibers, such as morphology and surface properties.

## 2. Materials and Methods

### 2.1. Chemicals and Reagents

Poly(ε-caprolactone) (PCL) was a CAPA^®^ 6800 homopolymer with purity of 99% and molecular weight of 80,000, purchased from Solvay Interox Ltd. (Warrington, UK), GB. α-tocopherol (αT) with purity ≥96% and density of 0.95 g/cm^3^, as model compound for vitamin E, 1,2-dichloroethane (DCE) (anhydrous, 99.8%), (2,2-diphenyl-1-picrylhydrazyl) (DPPH) stable free radical and 2,2′-azino-bis(3-ethylbenzothiazoline-6-sulfonic acid) diammonium salt (ABTS) were supplied by Sigma-Aldrich (St. Louis, MO, USA). All other chemicals and reagents were of pharmaceutical and analytical grade.

### 2.2. Preparation of PCL/α-Tocopherol Solutions

Poly(ε-caprolactone) and α-tocopherol were dissolved in 1,2-dichloroethane as common solvent, by continuous stirring overnight at room temperature. The solutions were prepared starting from PCL at the selected concentration of 20% (*wt*/*v*), which gave good electrospinning behavior in preliminary tests, with smooth, non-beaded fibers of round cross-section. α-tocopherol was added to the 20 wt% PCL solutions in various amounts of 1, 2, 5, 10 and 20% (*wt*/*wt*) in respect with PCL amount.

### 2.3. Rheology of PCL/α-Tocopherol Solutions

Rheological properties of the PCL/αT solutions were determined using a Physica MCR 301 rheometer (Anton Paar, Graz, Austria) equipped with cone-plate geometry of 50 mm diameter (CP50) and 99 μm truncation. A Peltier heating system was used for accurate temperature control at 25 ± 0.1 °C. Strain sweep experiments were preliminary performed at a frequency of 10 rad/s to determine the region of linear viscoelastic response. Oscillatory frequency sweeps were performed over the angular frequency range of 500–0.05 rad/s at the appropriate strain in the linear region to determine the storage (G′) and loss (G″) moduli. Rotational measurements were performed using shear rates from 0.1 to 1000 s^−1^ to determine the viscosity, which was calculated as
(1)$$\eta =\tau /\dot{\gamma },$$
where $\dot{\gamma }$ is the shear rate and *τ* is the shear stress. After loading each sample was hold for ~5 min before testing to allow stress relaxation and temperature equilibration.

### 2.4. Electrospinning

Electrospinning was carried out using a conventional horizontal set-up consisting of a syringe pump, a high voltage power source and a flat collector electrode covered with aluminum foil. The PCL/αT solutions were feed with a controlled syringe pump, at room temperature, from 5 mL glass syringes capped with metal capillary needles having inner diameter of 0.62 mm. A high voltage DC power supply (0–20 kV) was connected between the metal needle and the grounded collector electrode.

The deposition parameters (applied voltage, flow rate, distance between collector and needle) in combination with solution properties (concentration, conductivity, viscosity) have a great effect on the properties of obtained fibers. Throughout the trials, the feed rate was varied between 0.135 and 0.179 mL/min, the collection distance was varied from 10 to 20 cm and the voltage was varied up to 17 kV, values above 10 kV being generally noticed to be adequate in overcoming the surface tension of the solutions and in generating a Taylor cone. The processing parameters for the production of continuous, non-beaded PCL/αT fibers were applied voltage of 15 kV, distance between the spinneret (the tip of the needle) and the flat aluminum collector of 15 cm and flow rate of 0.179 mL/min [39].

### 2.5. ATR-FTIR Spectroscopy

A Bruker VERTEX 70 spectrometer, equipped with a single reflection diamond ATR crystal with an incidence angle of 45 °C was used for recording the infrared spectra of the PCL/αT fiber mats. The background and sample spectra were obtained by collecting 32 scans in the 600–4000 cm^−1^ wavenumber range, with a resolution of 4 cm^−1^. The fiber samples were folded for a better contact with the ATR crystal and pressed with a flat-tip plunger until spectra with suitable peaks were obtained. All experiments were performed in triplicate and represented as average spectra. The processing of spectra was achieved using OPUS 6.5 program.

### 2.6. Scanning Electron Microscopy (SEM)

To study the morphology of the obtained electrospun fibers a Leica Cambridge Stereoscan S360 scanning electron microscope was used in high vacuum mode, with an accelerating voltage of 20 kV. The fiber specimens were fixed on stubs and sputter-coated with gold prior to SEM imaging using an Edwards Sputter Coater S150B at 25–40 mA for 120 s. The average fiber diameter and diameter distribution were determined by randomly measuring the diameters of the fibers at minimum 100 different points from SEM images with Image J software. Magnification is indicated on micrographs.

### 2.7. Atomic Force Microscopy (AFM)

The topography of fibers’ surface was analyzed with a Solver Pro-M Scanning Probe Microscope (NT-MDT, Zelenograd, Russia) in atomic force microscopy (AFM) configuration, the images being acquired in air at room temperature. For each sample a surface of 2 × 2 μm^2^ was scanned. The surface roughness was determined by statistical estimation. The NT-MDT Nova v.1.26.0.1443 software was used for image acquisition and analysis.

### 2.8. Water Contact Angle Measurements (WCA)

The static water contact angles (WCA) for the PCL/αT electrospun fibers were determined by the sessile drop method, using a CAM-200 instrument from KSV—Helsinki, Finland at room temperature, within 10 s after placing 1 μL drops of water on the surface of the fibrous mats. Contact angle measurements were taken at least 5 times at different locations on the surface and the average values were reported.

### 2.9. DPPH Antioxidant Assay

Amounts of ~10 mg of electrospun fibers carefully detached from the aluminum foil were placed in vials containing 2 mL of methanol and were left shaking for 3 h at room temperature. 0.5 mL of supernatant from each sample were mixed with 2 mL of freshly prepared 2,2-diphenyl-1-picrylhydrazyl (DPPH) 0.06 mM solution in methanol, vigorously vortexed for 5 min, then left for 30 min at room temperature in the dark. The control was prepared adding 500 µL methanol to 2 mL DPPH solution. The radical scavenging activity of the PCL/αT fibers was evaluated based on the discoloration of purple DPPH solution, which reflects the amount of the remaining DPPH radicals in the solution, by measuring the absorbance at 517 nm using a Hewlett-Packard 8450A UV/Vis Spectrophotometer.

The radical scavenging activity (RSA) was calculated as the percentage of DPPH radical inhibition according to the equation:(2)$$RSA\%=(1-A_{sample}/A_{control})\times 100$$
where *A_sample_* represents the absorbance of the sample solution and *A_control_* represents the absorbance of the control solution. Each measurement was carried out in triplicate and data were expressed as average.

### 2.10. ABTS Antioxidant Assay

The green colored ABTS•^+^ free radical cation solution was obtained by mixing equal volumes of a 7.4 mM aqueous solution of 2,2′-azino-bis(3-ethylbenzothiazoline-6-sulfonic acid)diammonium salt (ABTS) and a 2.6 mM aqueous solution of potassium persulphate, then allowing them to react overnight in the dark at room temperature. Prior to assay, the obtained ABTS•^+^ solution was diluted with ethanol to an absorbance of ~0.77 ± 0.05 measured at 750 nm.

Amounts of ~8 mg of electrospun fibers from each sample were kept in 5 mL solution 95/5 ethanol/water (*v*/*v*); the vials were left shaking overnight at 23 °C and 120 rpm in a thermostated bath. 50 μL of supernatant from each sample or pure αT were added to 1 mL of the diluted ABTS•^+^ solution and the absorbance at 750 nm was read after 10 min incubation in the dark at room temperature. Control sample was prepared with 50 μL ethanol added to 1 mL ABTS•^+^ solution. The absorbance at 750 nm of each sample was read using a Cary 60 UV–Vis spectrophotometer (Agilent Technologies) and the *RSA%* inhibition of ABTS•^+^ was calculated with the same Equation (2) as for DPPH.

### 2.11. Migration Study of α-Tocopherol from PCL/αT Electrospun Fiber Coatings

α-Tocopherol migration from the PCL/αT fibers was investigated at room temperature, using 95% *v*/*v* aqueous ethanol solution as an alternative food simulant generally accepted for fatty foods [40,41]. This simulant was chosen for compliance testing in the framework of EU Regulations, considering the request of worst case foreseeable conditions for migration and the hydrophobic nature of the vitamin E [35]. Strips of fiber mats of ~2.5 cm × 2.5 cm from each sample were carefully detached from the aluminum foil, weighed and kept at dark for minimum 10 days in tightly closed vials containing 5 mL of release medium (95% ethanol). Sampling was carried out at given intervals (each 24 h) and the migration rate from the electrospun fibers was determined by UV-Vis spectroscopy using a Cary 60 UV-Vis spectrophotometer (Agilent Technologies, Santa Clara, CA, USA) by scanning from 200 to 600 nm. The absorbance readings were carried out at λ_max_ = 290 nm, as previously determined from the calibration curve. Samples were run in quartz cuvettes with 1 mm path length. The absorbance readings were replicated three times. Sample thickness was measured with a Digital Mikrometer DIN 863 Messzeuge (MIB) Germany in 5 points, taking the average value for calculations.

The migrant transfer from one phase to the other in a two-phase food (or food simulant)/polymer system occurs to reach the thermodynamic equilibrium. The dimensionless partition coefficient can be defined as the ratio of the migrant concentration in the sample (*C_sample_*_,∞_) to the migrant concentration in the food simulant system (*C_simulant_*_,∞_) at equilibrium:(3)$$K_{p}=C_{sample,\infty }/C_{simulant,\infty }$$

The *K_p_* partition coefficient is a measure of the chemical affinity of the migrant towards the sample or the food simulant. When *K_p_* = 1, the migrant concentration in the food simulant system equals the concentration in the sample at equilibrium. *K_p_* > 1 and *K_p_* < 1 describe a higher affinity of the migrant towards the sample and, respectively, towards the food simulant [42,43].

## 3. Results and Discussion

### 3.1. Rheological Behavior of PCL/α-Tocopherol Solutions

The **steady shear rate measurements** offer information on flow behavior of the solutions, which is essential in electrospinning processes.

The low molecular weight α-tocopherol showed low shear viscosity and typical Newtonian behavior, characterized by linear dependence of shear stress with shear rate (no variation of viscosity), over the entire tested range—Figure 1. PCL, with long macromolecular chain and high molecular weight, has much higher shear viscosity (up to ~700 Pa·s) and shear stress (up to ~70,000 Pa) in solutions, especially in concentrated ones such as 20 wt% in our case, when chain entanglements have strong effect on the viscoelastic behavior [44]. The solution of PCL presented a shear thinning behavior, due to disentanglement of macromolecular chains under higher shear rates. Addition of 1% α-tocopherol to PCL strongly decreased the viscosity and the shear stress of solution. This effect suggests the apparition of interactions, most probably of intermolecular hydrogen bonds between the C=O groups in PCL and the phenolic OH in α-tocopherol, which reduce the entanglement of PCL chains, thus decreasing the viscosity [45,46]. Further increase of α-tocopherol content had less effect on rheological behavior, the viscosity curves overlapping for shear rates above ~10 s^−1^. The shear thinning behavior was observed for all PCL/αT solutions, showing that sufficient entanglement density is maintained, suitable for production of uniform, non-beaded fibers through electrospinning [47]. The red square in Figure 1a shows the shear rate range equivalent with the calculated shear rate in our electrospinning process (see beginning of Section 3.2).

The **dynamic oscillatory measurements** allowed the evaluation of the viscoelastic response of the PCL/αT fiber-forming solutions. Figure 2 shows the dynamic moduli variation at different angular frequencies for the PCL solutions with various contents of α-tocopherol.

Figure 2 showed typical liquid behavior of α-tocopherol, with loss modulus, G″ (which describes the viscous character of material), being much higher than the storage modulus, G′. Rheology of PCL solution indicated an important contribution of elasticity, the values of G′ being very close to those of G″. However, the behavior is predominantly viscous (liquid-like), with G″ > G′ on almost entire frequency range. A crossover point appeared at ω = 123.6 rad/s, where G′ = G″ = 3695 Pa; this indicates a behavior of entangled polymer solution. The dynamic moduli decreased with increasing amounts of tocopherol in the PCL solutions, the effect being much stronger for the storage modulus, G′. Therefore, structuration of PCL macromolecular chains is reduced under the effect of tocopherol, as mentioned before, diminishing the elastic behavior of the solutions. Interesting to note is that if the addition of only 1 wt% α-tocopherol strongly decreased the viscosity (Figure 1a), the effect on G″ was rather low. Higher amounts of 10 and 20 wt% αT lead to similar behavior, the G′ and G″ curves almost overlapping. The increase of moduli (especially of G′) with ω became more pronounced with increasing the tocopherol amount.

**Thixotropy tests** were performed to investigate the time-dependent behavior, generally used to assess structural regeneration of solutions after passing through mechanical stress. A three step oscillation test (3ITT) was performed at constant angular frequency ω of 10 rad/s (which is in the linear viscoelastic range—LVE), as following. A low amplitude strain (γ) of 10% was applied for 150 s, to simulate the behavior at rest, after which γ was increased to 100% for 50 s, to simulate structural breakdown of the sample during high mechanical stress (e.g., in electrospinning processes). Recovery behavior was tested by returning to 10% amplitude strain for another 150 s.

The destructuration of PCL macromolecular chains under mechanical stress was reversible, the tan δ returning to the initial values by complete relaxation. Addition of low molecular weight tocopherol strongly diminished the capacity of the system to absorb and store energy during mechanical stress at high amplitude strains, the loss tangent at relaxation (after 200 s in Figure 3) being much higher than for PCL, mainly due to the decrease of the elastic modulus, G′, as previously discussed. Disruption of intermolecular junctions by hydrogen bonds is not totally reversible for low amounts of tocopherol (1, 2, 5 wt%), the final tan δ at relaxation (after 200 s) remaining at values higher than before application of high stress (before 150 s). The solutions with high amounts of 10 and 20 wt% tocopherol totally recovered the tan δ, similar with the solution containing only PCL. It is to note that the final tan δ was similar for all solutions containing tocopherol, regardless of concentration, while the initial tan δ was significantly higher when tocopherol was in high amounts (10 and 20 wt%). Therefore, the different recovery behavior is in fact due to the initial structuration in solutions. It appears that interactions through hydrogen bonds have strong influence on the structuration of PCL macromolecular chains mainly at low concentrations of tocopherol. However, these are counterbalanced at high concentrations of tocopherol, when the entanglement is enhanced due to the diminished solvation of PCL, a significant part of 1,2-dichloroethane solvent being involved in interactions with the vitamin E constituent. Zhang et al. [16] also observed non monotonous rheological behavior with α-tocopherol concentration in chitosan/zein systems, which indicated that high concentration of α-tocopherol led to inhibition of interactions, probably through electrostatic repulsion between molecules. Based on these observation only the solutions with 1, 5 and 20 wt% α-tocopherol were used for obtaining fibers by electrospinning.

### 3.2. Characterization of PCL/α-Tocopherol Electrospun Fibers

Generally, the shear rate of flow in capillaries, tubes and pipes is calculated using the Hagen-Poiseuille formula [28].
(4)$$\dot{\gamma }=4V/\pi R^{3}t$$
where $\dot{\gamma }$ represents the shear rate at the wall, *V*/*t* is the volumetric flow rate and *R* is the inner radius of the needle.

Based on the electrospinning parameters used in this study, the calculated value of the corresponding shear rate was of ~128 s^−1^. One can correlate this value with the shear range marked by the red square in Figure 1a, in which the viscosity of solutions was similar for all tocopherol concentrations above 1 wt%.

Fibers were obtained by electrospinning from the PCL/α-tocopherol solutions and the composition, morphology, surface properties, antioxidant activity were determined by ATR-FTIR spectroscopy, SEM/AFM microscopy, contact angle measurements and DPPH/ABTS assays. Migration studies in a food simulant were also performed.

**The ATR-FTIR spectra** in Figure 4 helped reveal the effect of α-tocopherol on PCL in fiber formation by electrospinning. The PCL fibers showed characteristic bands at 2942 and 2865 cm^−1^ (stretching vibrations of alkyl aliphatic C–H), a sharp band at 1724 cm^−1^ (stretching ester carbonyl C=O) and bands at ~1165–1105 cm^−1^ (stretching ester C–O). The presence of α-tocopherol in fibers was indicated by the small peak at 1090 cm^−1^, corresponding to the plane bending of phenyl ring. This was shifted to higher wave numbers from its position at 1083 cm^−1^ in the spectra of the pure compound, indicating a relaxation of the aromatic ring due to lower effect of the –OH hydroxyl group, which is involved in hydrogen bond interactions with the carbonyl groups in PCL. It is known the ability of phenolic OH to form strong H-bonds [48] and therefore to modify the conformation of the PCL macromolecular chains, limiting chain flexibility and mobility [45,46]. Similar hydrogen bond interactions were observed for α-tocopherol with the –OH hydroxyl groups of PVA [40]. The small peak at 3640 cm^−1^ in tocopherol disappeared in fibers but a new, larger peak appeared around 3670 cm^−1^, also indicating intermolecular interactions with PCL. These observations suggest that the interactions between α-tocopherol and PCL macromolecular chains in solutions, observed in the rheology behavior, are preserved in the solid fibers after evaporation of the solvent during the electrospinning process.

**The SEM microscopy** showed that fibers with thickness in micrometer domain, having general homogeneous morphological aspect of long, continuous, smooth and non-beaded threads were obtained by electrospinning from all PCL containing solutions—Figure 5. The fibers from PCL and PCL with 20 wt% α-tocopherol appeared similar, with threads of various diameters and with size variation along the length. The size histograms, obtained from the 200 μm magnification images, showed the same average diameter of 5.6 μm and similar size distribution in the range of 2–10 μm. This is in good agreement with the recovery behavior observed from the thixotropy tests, the solutions having sufficient molecular chain entanglement to prevent breaking of the electrically driven jet that allows formation of continuous fibers [49]. Small amounts of only 1 wt% α-tocopherol increased the uniformity of the fibers and narrowed the size distribution, the threads with the diameter close to the average size of 5.9 μm being dominant. It is known that lower viscosity coupled with enough entanglement density improves the filament stretching and promotes the formation of finer and smoother fibers [47,49]. In our case, the solution with small amounts of 1 wt% tocopherol had lower viscosity than the solution with only PCL, as shown in Figure 1a, explaining the observed results in SEM micrographs. Intermediate content of 5 wt% α-tocopherol produced thicker fibers with average diameter of 6.4 μm, with wider size distribution.

**The Atomic Force Microscopy** revealed the surface topography of the obtained electrospun microfibers. The 2D and 3D AFM micrographs in Figure 6 showed that low amounts of 1 and 5 wt% tocopherol had marginal effect on the roughness of the electrospun fibers, while high amounts (20 wt%) strongly increased the irregularities of the surface.

The height histograms in Figure 7 showed a shift towards smoother surface and narrower distributions for 5 and especially 1 wt% tocopherol; this is due to the rheological behavior of initial solutions, as previously discussed. Contrary, high amount of 20 wt% tocopherol had a strong effect on surface roughness, which has grown significantly and became bimodal.

**The contact angle measurements** in Figure 8 showed that the mats of electrospun fibers from PCL had low surface wettability, reflected by the high contact angle of 121°. Hydrophobicity was enhanced by the addition of α-tocopherol, small amounts of 1 wt% having the highest effect. Strong increase of tocopherol content from 5 to 20 wt% had only marginal effect on increasing the contact angle from 138.9 to 142.1°, and this could be due to increased roughness of the fibers, as shown by AFM micrographs in Figure 6 and the corresponding histograms in Figure 7.

Materials for food contact applications require specific wettability, which is regulated by surface free energy and surface structure [50]. It is well known that a low surface energy induce a hydrophobic character, with low wettability reflected in high contact angles. Electrospun fibers may represent an attractive alternative for wettability manipulation due to their large surface area to volume ratio, increased surface roughness and porous structures [51]. Highly hydrophobic surfaces are usually achieved by modifying rough surfaces with low-surface energy materials [52]. The values obtained for the water contact angles show that poly(ε-caprolactone) electrospun fibers containing α-tocopherol are highly hydrophobic, being suitable to induce water repellent properties in packaging materials. Considering that a network of fine intertwined fibers is formed by electrospinning but not a compact layer, the high apparent contact angle is mainly due to 3D structuration and to the entrapment of air within the structure. This could provide sufficient barrier properties for slow water penetration.

The **antioxidant activity** of the electrospun fibers was determined based on the radical scavenging reaction with DPPH• and ABTS•^+^, decreasing the absorbance in the visible region of the spectrum at the corresponding λ_max_ values. The DPPH and ABTS assays are two simple, low-cost, frequently used methods to evaluate the antioxidant activity of a wide range of natural polyphenolic compounds [40,53]. Both tests can be used to determine the antioxidant activity of hydrophilic and hydrophobic compounds.

It can be noticed from Figure 9 that α-tocopherol shows a very high radical scavenging activity, with instant inhibition of DPPH (due to the ability to donate the hydrogen atom from the –OH group onto the aromatic ring, with formation of tocopheroxyl radicals [54]) and, respectively, 85% inhibition for ABTS. Martins et al. [55] also reported close to 100% radical scavenging activities in DPPH test for chitosan films with 0.1 and 0.2% α-tocopherol. Differences between DPPH and ABTS results can be related with specific mechanism of action, namely hydrogen extraction for DPPH and single electron transfer for ABTS.

At the opposite side, PCL showed no antioxidant activity, as expected. Addition of tocopherol to PCL induced antioxidant activity of fibers. It can be noticed that at low α-tocopherol concentrations in the microfibers, the effectiveness of the hydrophobic additive is not high but when concentration increases most of the α-tocopherol is effective as antioxidant, especially for the neutralization of DPPH. A 20 wt% tocopherol content in fibers lead to total quenching of DPPH but only to 15.5% for ABTS radical. ABTS was much less affected, due to the different mechanism of action with the cation radical in hydrophobic media [56].

**Migration tests** in 95% ethanol as food simulant were carried out to investigate the release of the bioactive compound from PCL/α-tocopherol electrospun fibers.

The migration profiles of α-tocopherol from the electrospun fibers of PCL with α-tocopherol (Figure 10) showed a release behavior influenced by the initial amount incorporated in the fiber. On the investigated 13 days range, α-tocopherol was released in a relatively gradual manner but in a higher amount from PCL/1αT, reaching ~30% release from the initial amount. The samples with increased initial α-tocopherol amount showed faster release, reaching the plateau region in the first 24 h, but total migration was lower, of only 16.7 wt% for PCL/20αT. This behavior can be explained by formation of hydrogen bonds between carbonyl groups from PCL chains and phenolic OH groups in α-tocopherol, as previously discussed, revealing the higher affinity of the active component to the polymeric fibers than to the simulant. Fast migration in 95% ethanol was also reported by de Carvalho et al. [40] and Mirzaei-Mohkam et al. [41] when α-tocopherol was incorporated into poly (vinyl alcohol) or carboxymethyl cellulose films through nanoparticle carriers. They observed that increased loading of α-tocopherol leaded to higher initial rates but lower total release in plateau region, similar with our observation.

The fast and early release of α-tocopherol explains the antioxidant activity for DPPH and ABTS observed in Figure 9 and may inhibit microbial growth and lipid oxidation at initial storage stages of food [41,57].

The partition coefficient values in Table 1 are higher than unit (*K_p_* > 1), as also observed by Otero-Pazos et al. [58], corresponding to a higher affinity of the migrant towards the electrospun polymeric fibers. This kind of behavior is assigned as “negative migration”, which is preferred for systems that are expected to keep the active agent within the food packaging [59]. The overall migration levels (C), which represent the amount of migrant compound per mass unit of food simulant, were between 2.89 mg/kg for PCL/1αT and 26.2 mg/kg for PCL/20αT. All obtained values are much lower (below half) than the overall migration limit of 60 mg/kg food simulant, established for food contact packaging materials by the current European legislation [35].

**The respirometry tests** performed on an ER12 ECHO Instruments Respirometer showed a linear increase of oxygen consumption during the storage of curd cheese (blank in Figure 11), as a result of microbial growth that leads to food spoilage. This was slightly diminished by the PCL fibers, which maintained the linear evolution. Addition of α-tocopherol strongly inhibited the oxygen consumption, especially in the first 48 h, when the values were lower than for PCL alone. Therefore, the PCL/α-tocopherol electrospun fibers show retarding effect on microbial growth on cheese, delaying the food spoilage [55,60].

## 4. Conclusions

α-tocopherol was successfully embedded into poly(ε-caprolactone) biodegradable polymer by electrospinning to form antioxidant and antimicrobial fibers. Rheological behavior indicates interactions between the two components, strongly affecting the elastic behavior of PCL, especially at low concentrations of α-tocopherol. The structuration of macromolecular chains recovered after the shear stress, explaining facile formation of continuous, non-beaded fibers by electrospinning. ATR-FTIR spectra revealed the stabilization through H-bonding of the bioactive compound in the electrospun PCL fibers. A 3D construct was obtained, with a generally homogeneous morphological aspect and fibers diameters in the micro-scale range (~6 μm). Low amounts of α-tocopherol (up to 5 wt%) had only marginal effect on the topography of the fibers, while roughness strongly increased for 20 wt% αT content. Adding the vitamin E component to the PCL fibers induced a highly hydrophobic character and a strong antioxidant activity against DPPH and ABTS radicals. A significant reduction of microbial growth on cheese was observed by respirometry when the electrospun fibers were used in direct contact with the food. This was explained by the observed fast release of the active compound in food simulants. These outcomes indicate the potential use of electrospun fibers from PCL with incorporated α-tocopherol as bioactive materials for food packaging applications.

## Figures and Tables

**Figure 1 molecules-26-05498-f001:**
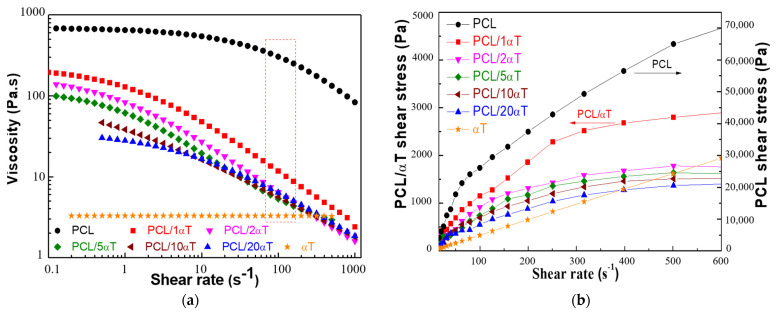
Viscosity (**a**) and shear stress (**b**) as function of shear rate for the PCL solutions containing α-tocopherol.

**Figure 2 molecules-26-05498-f002:**
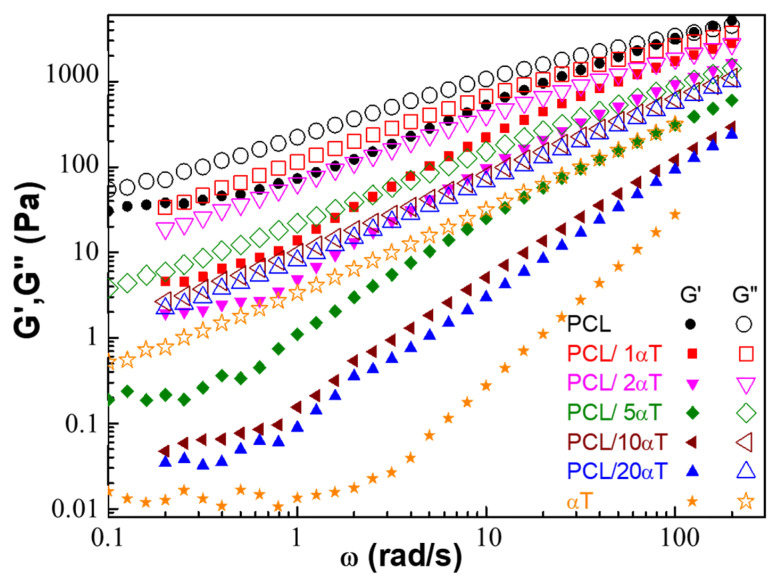
Angular frequency dependence of storage (G′, filled symbols) and loss (G″, open symbols) moduli for the PCL solutions containing α-tocopherol.

**Figure 3 molecules-26-05498-f003:**
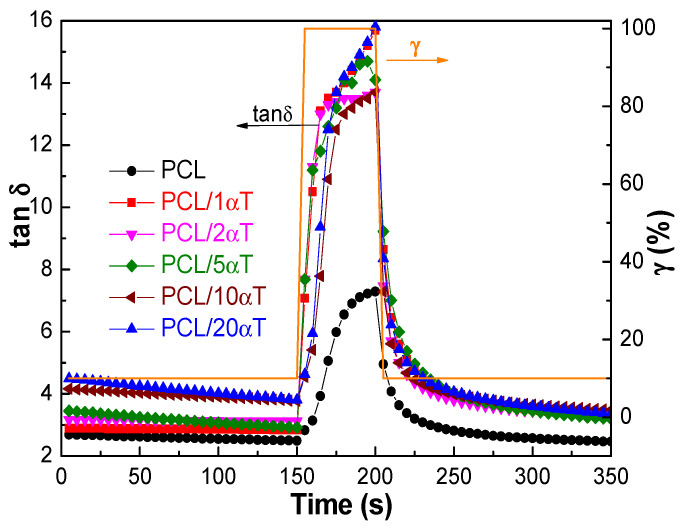
Time-dependent variation of the loss tangent, tan δ, for the PCL solutions containing α-tocopherol.

**Figure 4 molecules-26-05498-f004:**
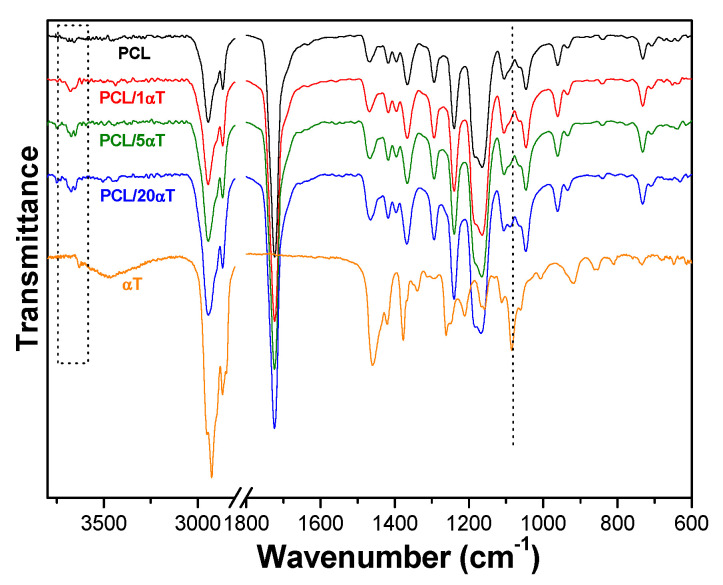
ATR-FTIR spectra of the PCL solutions with various α-tocopherol content.

**Figure 5 molecules-26-05498-f005:**
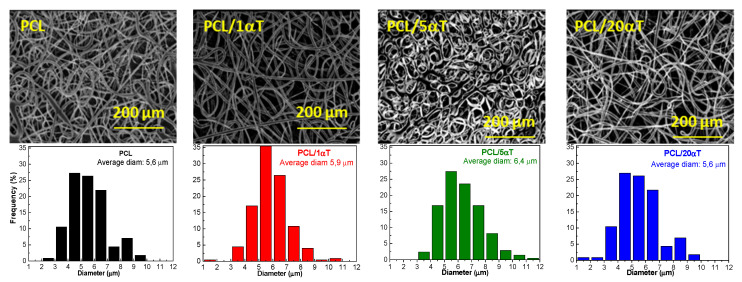
SEM images and fiber diameter distributions of electrospun fibers of PCL with 1, 5 and 20 wt% αtocopherol content.

**Figure 6 molecules-26-05498-f006:**
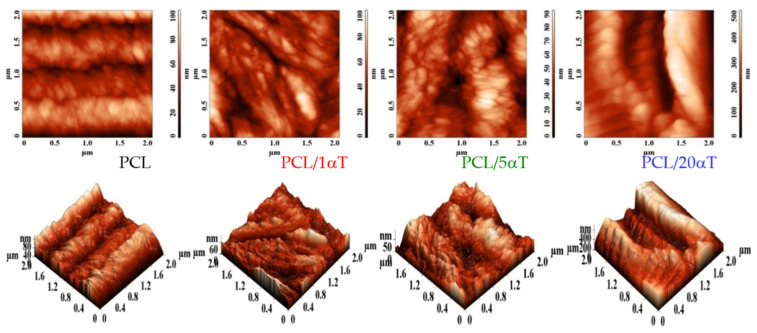
2D and 3D AFM images at 2 × 2 μm^2^ scan area of electrospun fibers of PCL with different α-tocopherol concentrations.

**Figure 7 molecules-26-05498-f007:**
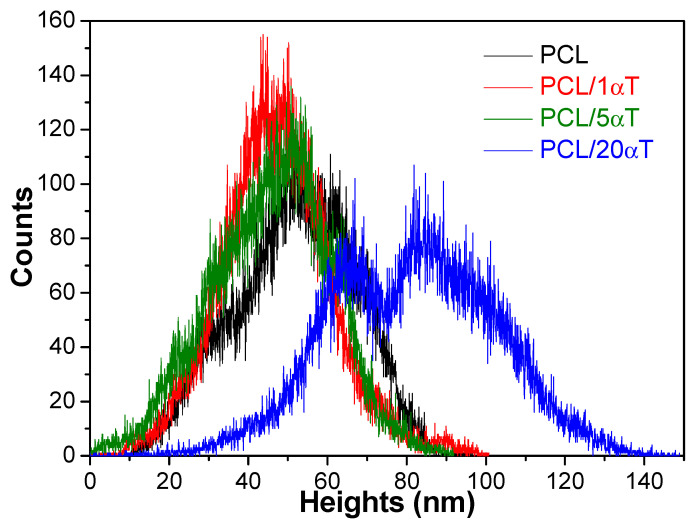
The heights histograms from the AFM micrographs of PCL/αT fibers.

**Figure 8 molecules-26-05498-f008:**
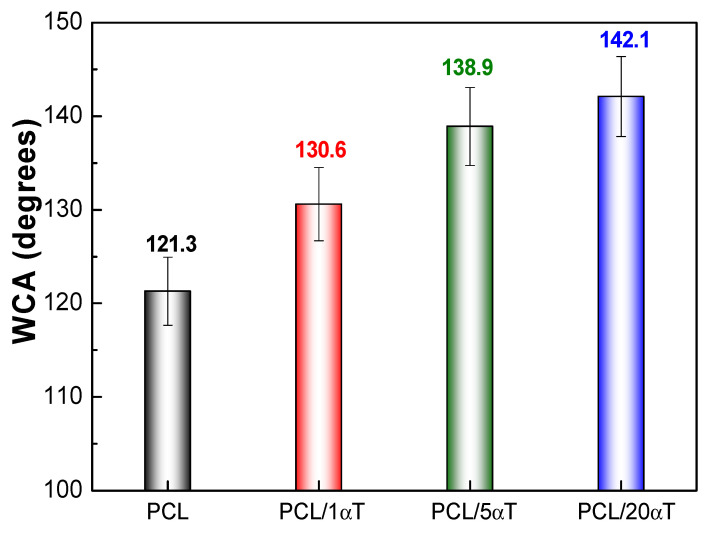
Water contact angle of PCL and PCL/αT samples.

**Figure 9 molecules-26-05498-f009:**
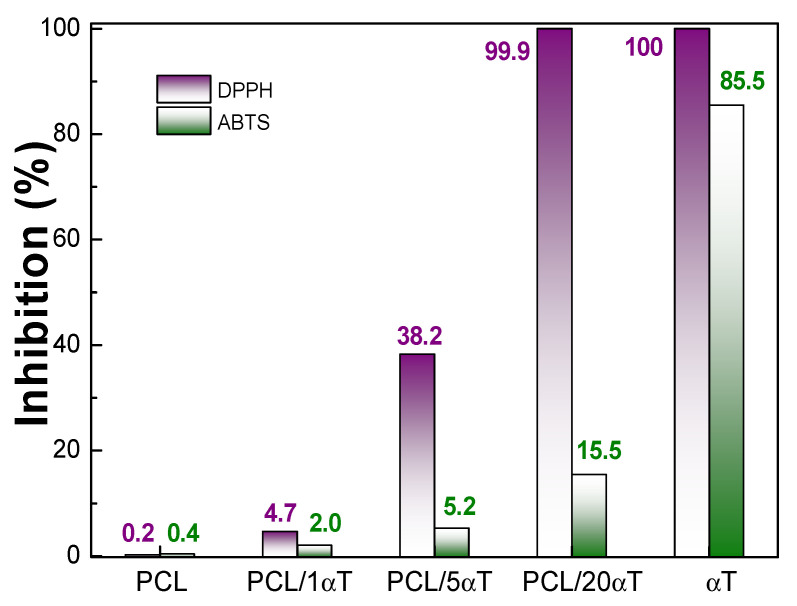
Radical scavenging activity of PCL/αT electrospun fibers.

**Figure 10 molecules-26-05498-f010:**
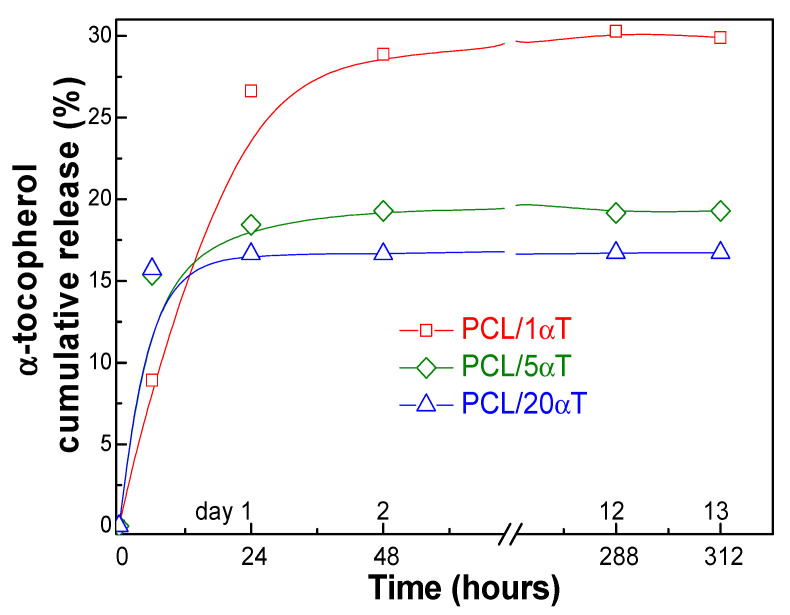
Migration of α-tocopherol from electrospun PCL/αT fibers into 95% ethanol.

**Figure 11 molecules-26-05498-f011:**
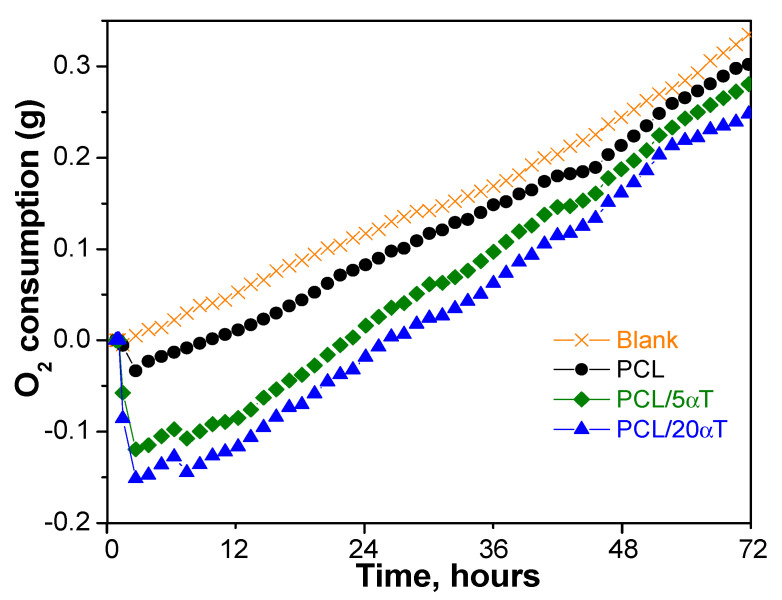
Oxygen consumption during curd cheese storage in presence of electrospun fibers from PCL/α-tocopherol.

**Table 1 molecules-26-05498-t001:** Migration parameters of α-tocopherol from PCL/αT electrospun fibers.

Sample	αT Migration, %	*K_p_*	C (mg/kg Food Sim)
PCL/1αT	29.89	2.345	2.89
PCL/5αT	20.14	3.965	9.09
PCL/20αT	16.70	4.988	26.22

## Data Availability

Not applicable.

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
