# Peer review of "Bioactive Electrospun Fibers of Poly(ε-Caprolactone) Incorporating α-Tocopherol for Food Packaging Applications"

_molecules, 2021, doi:10.3390/molecules26185498_

Round 1
Reviewer 1 Report
To clearly discriminate the presented submission from your previous conference paper:
Dumitriu, R.P.; Mitchell, G.R.; Davis, F.J.; Vasile, C. Functionalized Coatings by Electrospinning for Anti-oxidant Food Packaging. Procedia Manuf. 2017, 12, 59–65, doi:10.1016/J.PROMFG.2017.08.008
Please rewrite the sections as follows:
- Title,
- Abstract
- Conclusions
- Acknowledgments.
- References.
In the present form, the highlighted sections clearly suggest auto plagiarism and discredit authors and their achievements.
Author Response
Title was replaced with a new one. Changes added in abstract and conclusions were marked in red. Acknowledgement and references are totally different to those in Procedia Manuf. 2017, 12, 59-65.
Reviewer 2 Report
The paper presented the study on the fabrication of electrospun PCL fiber mats incorporated with bioactive antioxidant compound-tocopherol for food packaging application.The rheological behavior of the solution and the properties of the electrospun PCL was comprehensively studied. It could be considered to be published in the journal after some improvements. The comments are given below:
1. It is suggested to introduce more work on the advance in application of the electrospun fibers in food package.
2. It seems to be inappropriately to use fiber ‘mesh’ in the work as no special technique was described to prepare the fiber ‘mesh‘. If it is, how much is the mesh size?
Author Response
A new paragraph on recent advances in electrospinning for food packaging was added in Introduction section; this is marked in red. The term “mesh” was removed from the text.
Round 2
Reviewer 1 Report
The authors made the suggested improvements.